## RESEARCH ARTICLE

# Saccharomyces cerevisiae malate dehydrogenase Mdh1p lacking mitochondrial targeting signal can be re-localized to peroxisomes

Chutima Chan[1], Naraporn Sirinonthanawech[1], Brian K. Sato[2], James E. Wilhelm[3] and Chalongrat Noree[1,*]

## ABSTRACT

Yeast mitochondrial malate dehydrogenase, Mdh1p, is known to form supramolecular complexes with other tricarboxylic acid (TCA) cycle and mitochondrial dehydrogenase enzymes, including the aldehyde dehydrogenase, Ald4p. These complexes have been proposed to facilitate NADH channeling. Here, we demonstrate that in cells grown to saturation and stationary phases, the endogenous Mdh1p, expressed without its mitochondrial targeting signal (MTS), stays outside mitochondria, in both a diffuse cytoplasmic distribution and localized to distinct puncta. The puncta formed by MTS-lacking Mdh1p show no co-localization with the MTS-lacking Ald4p, suggesting that they do not co-assemble into a supramolecular complex in the cytoplasm. However, we found that the MTS-lacking Mdh1p does co-localize with its cytoplasmic counterpart, Mdh2p, in puncta. Interestingly, Mdh2p has recently been reported to form heterocomplexes with the peroxisomal Mdh3p and to be transported into peroxisomes to assist in the glyoxylate cycle. We also show that the MTS-lacking Mdh1p co-localizes with a fluorescent peroxisome marker, Pex3p. Our findings suggest that different malate dehydrogenases can enter peroxisomes, potentially as a means to make the glyoxylate pathway more efficient.

KEY WORDS: Malate dehydrogenase, Supramolecular assembly, Mitochondria, Peroxisome, Yeast

## INTRODUCTION

Malate dehydrogenase plays an important role in metabolism, catalyzing the NAD-dependent conversion of malate to oxaloacetate during the tricarboxylic acid (TCA) cycle. In yeast, there are three different isoenzymes of malate dehydrogenases, Mdh1p, Mdh2p, and Mdh3p, which are encoded by *MDH1*, *MDH2*, *MDH3*, respectively. Mdh1p contains a mitochondrial targeting signal (MTS) at the N-terminus consistent with it being a mitochondrial resident protein (McAlister-Henn and Thompson, 1987; Thompson et al., 1988). A peroxisomal targeting sequence type 1 (PTS1) is located at the C-terminus of Mdh3p, resulting in

peroxisomal localization (Steffan and McAlister-Henn, 1992). Mdh2p has no MTS or PTS, thus it stays in the cytoplasm (Minard and McAlister-Henn, 1991).

When cells are aerobically cultured with glucose as the primary carbon source, glycolysis produces pyruvate, which is subsequently converted to acetyl-CoA. Acetyl-CoA is used as a substrate in the TCA cycle, producing high energy electron carriers, including NADH and $FADH_2$. These molecules eventually enter the oxidative phosphorylation pathway to produce the energy currency molecule adenosine triphosphate (ATP), which is required for driving several biochemical reactions and biological processes. Under aerobic and nutrient-rich conditions, mitochondrial Mdh1p is a major contributor to the TCA cycle. In addition, mitochondrial Mdh1p as well as its cytoplasmic isoenzyme, Mdh2p, are known to help maintain a physiological balance of $NAD^+$/NADH within the mitochondria and cytoplasm. Although $NAD^+$ and NADH cannot be transported in or out of mitochondria, cells are able to use Mdh1p to convert malate to oxaloacetate (producing NADH from $NAD^+$), allowing mitochondrial aspartate aminotransferase, Aat1p, to synthesize aspartate from oxaloacetate. Aspartate is then transported into the cytoplasm, where cytoplasmic Aat2p transforms it into oxaloacetate. Finally, Mdh2p is responsible for making malate from oxaloacetate, which is coupled to the production of $NAD^+$ from NADH. By this means, the homeostasis of $NAD^+$ and NADH inside the cells is regulated (Bakker et al., 2001).

Under glucose-depleted conditions, or when other biomolecules (e.g. fatty acids) are supplied for yeast cell culture, peroxisomal Mdh3p is primarily used for the conversion of malate to oxaloacetate in the peroxisomal glyoxylate cycle. First, fatty acids are broken down via β-oxidation, producing acetyl-CoA that is then used in the glyoxylate cycle. Upon renewed availability of external nutrients, glycolysis and TCA cycle are preferentially employed as the catabolic processes for highly efficient energy production (Chew et al., 2019).

A previous study using native polyacrylamide gel electrophoresis and mass spectrometry demonstrated that several mitochondrial enzymes, including NADH dehydrogenases and certain enzymes in the TCA cycle, can assemble into supramolecular complexes in mitochondria to promote channeling of NADH and other metabolites (Grandier-Vazeille et al., 2001). As aldehyde dehydrogenase, Ald4p, and malate dehydrogenase, Mdh1p, were present in these complexes, we were curious as to whether these two enzymes directly interacted in the absence of other components of the supramolecular complex. Taking advantage of the well-defined MTS sequences of Ald4p and Mdh1p (Thompson et al., 1988; Noree, 2018; Noree and Sirinonthanawech, 2020), we removed the MTS coding sequence from the endogenous *MDH1* gene and introduced the coding sequence for the fluorescent protein mCherry downstream of the *MDH1* gene to enable the visualization of the modified Mdh1p protein. We also constructed yeast cells expressing Mdh1p(noMTS)-mCherry and Ald4p(noMTS)-GFP to observe whether co-localization occurs.

[1]Institute of Molecular Biosciences, Mahidol University, 25/25 Phuttamonthon 4 Road, Salaya, Phuttamonthon, Nakhon Pathom 73170, Thailand. [2]Department of Molecular Biology and Biochemistry, Charlie Dunlop School of Biological Sciences, University of California, Irvine, 2238 McGaugh Hall, Irvine, CA 92697, USA. [3]Department of Cell and Developmental Biology, School of Biological Sciences, University of California, San Diego, 9500 Gilman Drive (MC0347), La Jolla, CA 92093-0347, USA.

*Author for correspondence (chalongrat.nor@mahidol.edu)

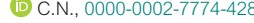 C.N., 0000-0002-7774-4281

Biology Open

Moreover, a recent study reported that a portion of Mdh2p, which is predominantly in the cytoplasm, can form heterocomplexes with peroxisomal Mdh3p, and these heterocomplexes can be transported into peroxisomes in a Pex5p-dependent fashion. As Mdh3p contains a PTS1 sequence that can be recognized by Pex5p (Klein et al., 2001), this likely facilitates the transport of Mdh2p/Mdh3p complexes into peroxisomes to enhance the efficiency of the glyoxylate cycle (Gabay-Maskit et al., 2020). Here, we examined whether other types of malate dehydrogenase heterocomplexes could have altered compartmentalization. By removing the mitochondrial targeting sequence from Mdh1p, we found that it co-localizes with Mdh2p in distinct puncta, suggesting they might form supramolecular complexes. We then found that Mdh1p co-localizes with the peroxisomal marker, Pex3p, suggesting that it is transported into peroxisomes. This suggests that different isoenzymes of malate dehydrogenases can be engineered to form heterocomplexes that can be targeted to peroxisomes, thereby opening the door to novel approaches to metabolic engineering with applications to biotechnology.

## RESULTS AND DISCUSSION
### MTS-lacking Mdh1p is able to form visible puncta outside mitochondria

Several mitochondrial enzymes have been reported to be able to form supramolecular structures (Misonou et al., 2014; Nasalingkhan et al., 2023, 2024; Noree et al., 2019). Mitochondrial malate dehydrogenase (Mdh1p) has been found in a supramolecular complex with several NADH dehydrogenases and TCA cycle enzymes (Grandier-Vazeille et al., 2001). To define what the requirements are for the formation of these supramolecular complexes, we used a strategy that had previously been used in the study of the filament-forming properties of the mitochondrial aldehyde dehydrogenase, Ald4p. When the MTS was removed from Ald4p, it formed filaments in the cytoplasm, suggesting that Ald4p filament formation was not dependent on the mitochondrial environment (Noree, 2018). We extended this approach by removing the MTS from Mdh1p and assaying its ability to form structures outside of the mitochondria (Fig. 1A). We found that in log-phase cells expressing Mdh1p(noMTS)-mCherry, the engineered Mdh1p molecules were still transported into the mitochondria. We could see the fluorescent signal forming a faint pattern of mitochondrial networks, with a portion of the enzyme found in dense clusters (Fig. 1B, upper left panel). This was similar to the mitochondrial pattern found in yeast expressing Mdh1p-mCherry (Fig. 1B, upper right panel). This was consistent with a previous study showing that MTS-lacking Mdh1p was still associated with the mitochondria as assayed by subcellular fractionation of log-phase yeast cells (Small and McAlister-Henn, 1997). Our results and the previous biochemical analysis suggest that MTS-lacking Mdh1p can still enter the mitochondria, perhaps via a piggybacking mechanism (where a protein without its own transport signal is carried to its destination by another protein) (Travers, 1999), mediated by some unknown mitochondrial component(s), enabling it to function in the TCA cycle. This is in agreement with the results of an *in vitro* import assay, that found only the pre-mature form of Mdh1p (derived from *in vitro* translation of an *MDH1* construct without MTS), can efficiently be imported into the isolated competent yeast mitochondria (Thompson and McAlister-Henn, 1989). Therefore, the internalization of the MTS-lacking Mdh1p into mitochondria might occur through piggybacking, similar to the way Mdh2p uses PTS-containing Mdh3p to get into the peroxisomes (Gabay-Maskit et al., 2020).

When cells were cultured to saturation and stationary phases (1-day and 5-day cultures, respectively), the distribution of Mdh1p(noMTS)-mCherry was primarily cytoplasmic (Fig. 1B, middle and lower left panels), whereas no cytoplasmic signal could be observed in yeast cells expressing Mdh1p-mCherry (Fig. 1B, middle and lower right panels). An assembly frequency analysis of the MTS-lacking Mdh1p (tagged with mCherry) in a yeast construct is detailed in Table S2. The analysis found that Mdh1p(noMTS)-mCherry structures were present in nearly 100% of cells across all three growth stages: log phase, saturation phase (1-day culture), and stationary phase (5-day culture). These frequencies were determined by counting a total of 251 to 291 cells for each condition. The results were highly consistent, with the average percentage of cells containing the assemblies being 100% for all conditions except for one clone in the stationary phase, which was 99.87%. This suggests that the metabolic state of the yeast cell affects the localization of Mdh1p. One possible explanation for this change is that the depletion of glucose in saturation and stationary phase cultures could alter the ability of Mdh1p to form supramolecular complexes. It could also be that the unknown component(s) required for assisting the translocation of the MTS-lacking Mdh1p into mitochondria are abundant during log phase, but less in other stages of growth. Based on our data that MTS-lacking Mdh1p showed extramitochondrial assembly when cells were grown to saturation and stationary phases, all following experiments were performed with cells cultured for 1 or 5 days (representing saturation phase and stationary phase, respectively).

### MTS-lacking Mdh1p and MTS-lacking Ald4p cannot co-assemble

Since both Mdh1p and Ald4p were previously reported to be components of supramolecular multi-enzyme complexes formed by TCA cycle and dehydrogenase enzymes, we sought to determine whether Mdh1p and Ald4p have a direct supramolecular interaction in the absence of other complex components. To address this, we created a yeast strain expressing Ald4p(noMTS)-GFP and Mdh1p(noMTS)-mCherry (Fig. 2A). As shown in Fig. 2B, despite observing the assembly of distinct Ald4p(noMTS)-GFP and Mdh1p(noMTS)-mCherry structures, no co-localization was evident. A co-localization frequency analysis of MTS-lacking Ald4p (tagged with GFP) and MTS-lacking Mdh1p (tagged with mCherry) was performed on a yeast construct during the stationary phase (5-day culture). The analysis found that there was 0% co-localization between the two proteins. The results showed that for Clone #1, an average of 65.26% of cells contained both protein structures, while for Clone #2, this average was 67.50%. A total of 309 to 332 cells were counted for each clone across the three experiments to determine these frequencies. Despite both proteins being present within the same cells, the analysis consistently found no co-localization across all experimental repeats. The analysis also found that for Clone #1, 18.23% of cells contained either structure, and 16.51% had no structures. For Clone #2, 16.20% of cells contained either structure, with 16.30% having no structures. The complete data are detailed in Table S3. This suggests that Ald4p and Mdh1p do not form heterocomplexes in the absence of other TCA cycle and mitochondrial dehydrogenase enzymes, implying that the supramolecular heterocomplex formation (proposed to support the channeling of metabolites and NADH molecules) requires the presence of other complex enzymes (Grandier-Vazeille et al., 2001).

### Co-localization of MTS-lacking Mdh1p and cytosolic Mdh2p suggests heterocomplex formation

A 2020 study reported that while Mdh2p lacks both an MTS (present in mitochondrial Mdh1p) or PTS (present in peroxisomal

Biology Open

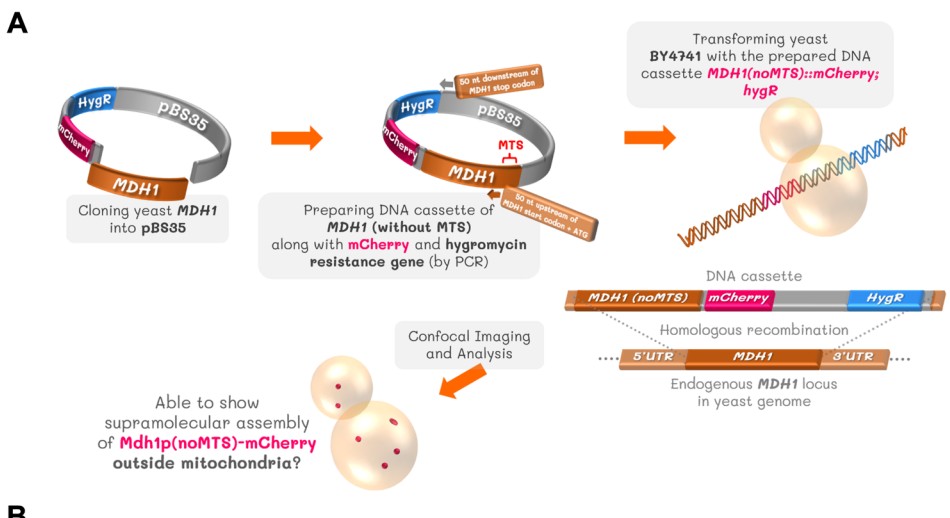

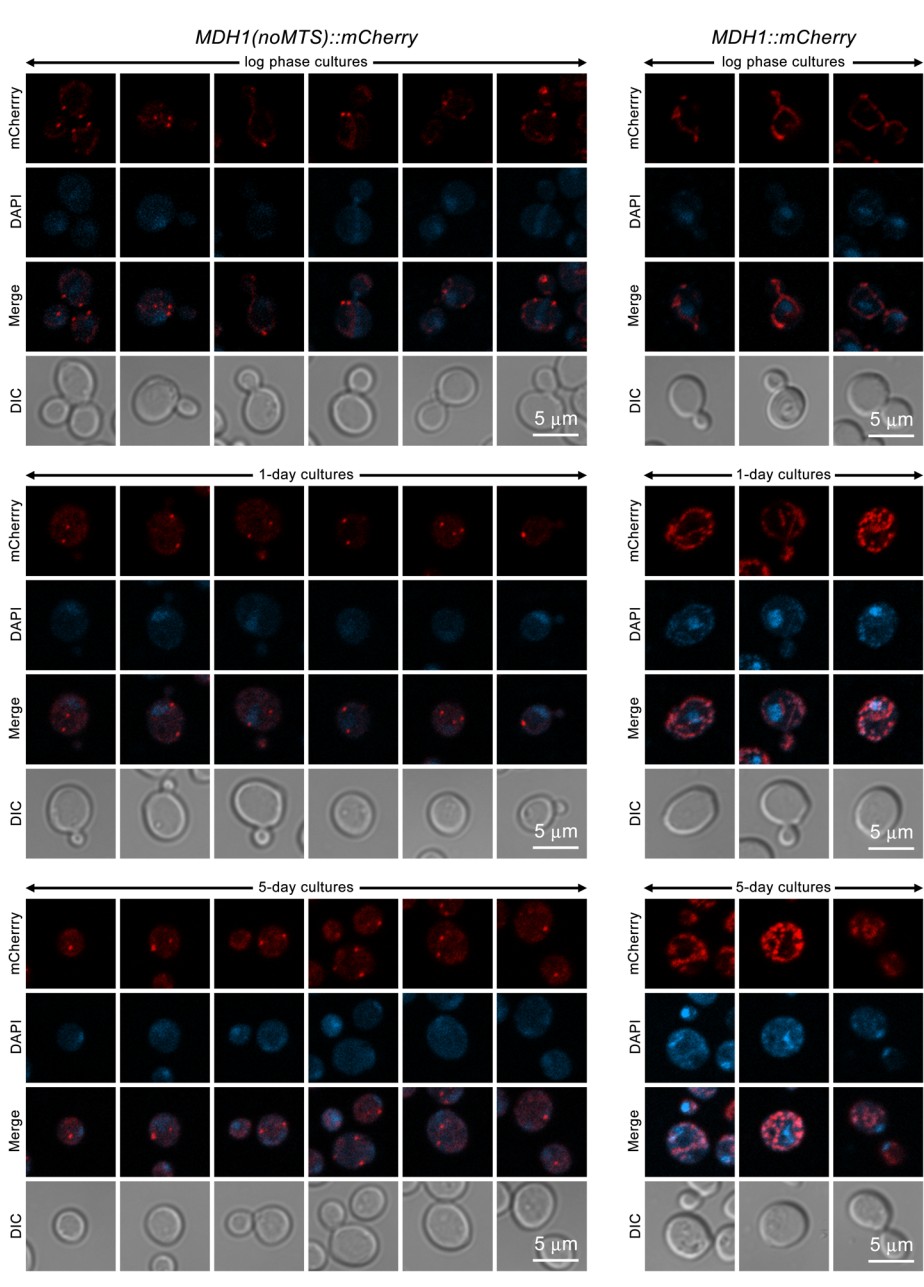

**Fig. 1. MTS-lacking Mdh1p localized to mitochondria during log-phase but was restricted to the cytoplasm in saturation and stationary phases.** A portion of the enzyme can assemble into visible foci. The *MDH1* gene was cloned into pBS35, and the resulting plasmid 'pBS35-MDH1' was used as a template DNA for preparing the DNA cassette by PCR. Yeast BY4741 was then transformed with the prepared DNA cassette. Through homologous recombination process, *MDH1* coding sequence without its MTS sequence along with mCherry coding sequence and hygromycin resistance gene can replace the original *MDH1* gene in the genome of base yeast strain BY4741 for checking whether the enzyme can form supramolecular structures outside mitochondria (A). Representative confocal images show two different clones of yeast expressing Mdh1p(noMTS)-mCherry (left panels) and two different clones expressing Mdh1p-mCherry (right panels, for comparison). The cells were cultured in liquid YPD to log-phase, saturation, and stationary phases. Fixed cells from each yeast clone were captured as Z-stack images from five different fields of view, each containing more than 50 cells. These Z-stack images were then compressed into two-dimensional images using maximum projection. Three independent experiments were performed with similar results (B). An assembly frequency analysis of the MTS-lacking Mdh1p (tagged with mCherry) revealed that structures were present in nearly 100% of cells across all three growth stages (log phase, saturation, and stationary phases). These frequencies were determined by counting a total of 251 to 291 cells for each experimental condition. The detailed data can be found in Table S2. Mdh1p, yeast mitochondrial malate dehydrogenase I; MTS, mitochondrial targeting sequence; noMTS, without mitochondrial targeting sequence; YPD, yeast rich culture medium containing yeast extract, peptone, and dextrose.

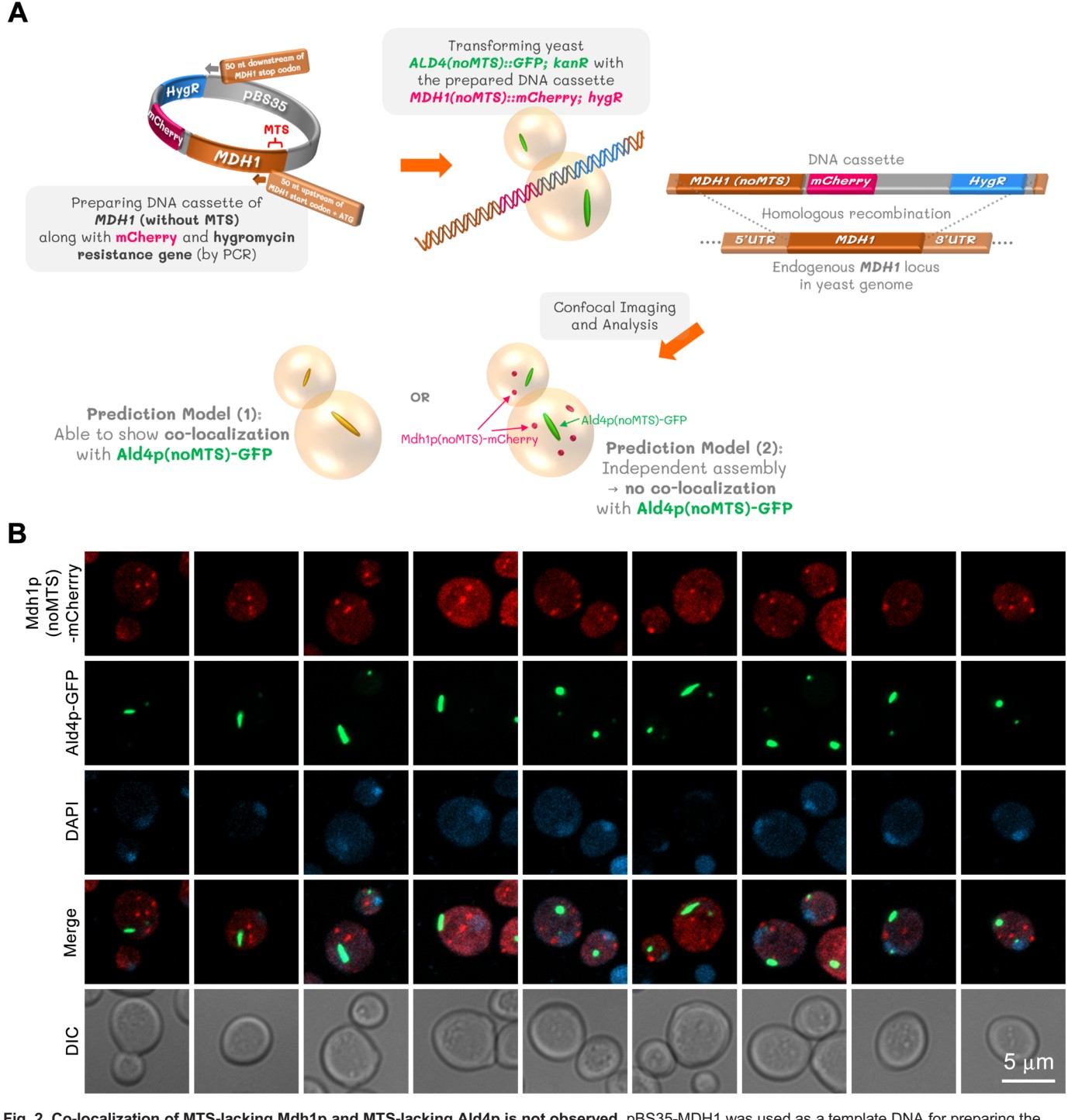

**Fig. 2. Co-localization of MTS-lacking Mdh1p and MTS-lacking Ald4p is not observed.** pBS35-MDH1 was used as a template DNA for preparing the DNA cassette by PCR. A yeast strain previously genetically engineered to express Ald4p(noMTS)-GFP was transformed with the prepared DNA cassette. The resulting yeast construct after verified to have the original *MDH1* removed from the genome and replaced by a version of *MDH1* without MTS sequence along with mCherry coding sequence and hygromycin resistance gene was subjected to imaging and analysis (A). Representative confocal images show two different clones of a verified yeast construct co-expressing Ald4p(noMTS)-GFP and Mdh1p(noMTS)-mCherry. The cells were cultured in liquid YPD to stationary phase (5-day cultures). Fixed cells from each yeast clone were captured as Z-stack images, which were then compressed into two-dimensional images using maximum projection. Three independent experiments were performed with similar results (B). A co-localization frequency analysis, which was performed on cells from the stationary phase (5-day cultures), showed that while an average of 65.26% (Clone #1) and 67.50% (Clone #2) of cells contained both protein structures, there was 0% co-localization between them. These frequencies were determined by counting a total of 309 to 332 cells for each clone. Detailed data can be found in Table S3. Ald4p, yeast mitochondrial aldehyde dehydrogenase IV; GFP, green fluorescent protein).

Mdh3p), it localizes to the peroxisome because a portion of Mdh2p can form a heterocomplex with Mdh3p. It has also been demonstrated that the co-transport of Mdh2p and Mdh3p into the peroxisome is facilitated by Pex5p, a peroxisomal membrane receptor capable of recognizing PTS1-containing proteins, through a mechanism termed 'piggybacking' (Gabay-Maskit

et al., 2020). This led us to question whether MTS-lacking Mdh1p can also form heterocomplexes with Mdh2p and Mdh3p. Since Mdh3p has its PTS1 sequence at the C-terminus, and C-terminally GFP-tagged Mdh3p (Mdh3p-GFP fusion protein) disrupts PTS1 recognition by Pex5p (Gabay-Maskit et al., 2020), we decided to test heterocomplex formation between Mdh1p(noMTS)-mCherry and Mdh2p-GFP instead (Fig. 3A). In yeast expressing Mdh2p-GFP and Mdh1p(noMTS)-mCherry, we observed the co-localization of MTS-lacking Mdh1p and Mdh2p (Fig. 3B).

A co-localization frequency analysis of MTS-lacking Mdh1p (tagged with mCherry) and Mdh2p (tagged with GFP) was performed during the saturation phase (1-day culture). The results showed that for Clone #1, an average of 71.27% of cells contained both Mdh1p(noMTS)-mCherry and Mdh2p-GFP structures, while for Clone #2, this average was 70.59%. A total of 152 to 164 cells were counted for each clone across the three experiments to determine these frequencies. Within these cells, the analysis found that the two proteins displayed 100% co-localization. The analysis also found that for Clone #1, 19.39% of cells showed either structure and 9.33% had no structures. For Clone #2, 20.17% of cells showed either structure, with 9.24% having no structures. The complete data are detailed in Table S4.

Using AlphaFold2 for structural modeling and template modeling (TM)-align for superimposition, we found a root mean square deviation (RMSD) of 1.69 and a TM-score of 0.91 (Fig. 4A-C). A TM-score, which ranges from 0 to 1 (with 1 being a perfect match), more accurately assesses global structural similarity than RMSD. Scores above 0.5 typically indicate similar folds, and our result points to a high degree of structural conservation. An RMSD below 2 Å also suggests significant structural similarity. This compatibility in structure indicates that Mdh1p and Mdh2p are likely capable of associating into larger complexes.

For comparison, we also investigated the structural relationship between Mdh2p and Mdh3p, as well as that between Mdh1p and Mdh3p. Mdh2p and Mdh3p showed an RMSD of 1.96 and a TM-score of 0.84 (Fig. 4D-F), while Mdh1p and Mdh3p exhibited a slightly higher sequence identity of 50% and an even closer structural alignment, with an RMSD of 1.52 and a TM-score of 0.91 (Fig. 4G-I). However, we chose Mdh2p for co-assembly experiments because tagging Mdh3p at its C-terminus could disrupt its peroxisomal targeting due to the presence of a PTS1 motif.

Collectively, these findings suggest that Mdh1p(noMTS) and Mdh2p possess sufficient structural compatibility to enable their association. This supports the broader concept that different malate dehydrogenase isozymes, despite their evolutionary paths, maintain enough structural similarity to form supramolecular assemblies even when outside their usual organellar locations.

Thus, our work, consistent with findings from others (Gabay-Maskit et al., 2020), indicates that heterocomplexes may be formed among different isozymes of malate dehydrogenases (Mdh1p/Mdh2p and Mdh2p/Mdh3p). As there is currently no report of Mdh2p localization to the mitochondria, it is unlikely that MTS-lacking Mdh1p enters the mitochondria via piggybacking on Mdh2p. Instead, its mitochondrial import may involve competition between Mdh2p and another mitochondrial protein. Further studies are necessary to elucidate the mechanism by which MTS-lacking Mdh1p enters the mitochondria during log-phase, identify the molecule(s) mediating this translocation, and determine why the internalization of MTS-lacking Mdh1p into mitochondria is terminated during saturation and stationary phases.

## MTS-lacking Mdh1p co-localizes with a peroxisomal marker, suggesting translocation

A portion of cytosolic Mdh2p has been suggested to enter the peroxisome through its interaction with Mdh3p. As we observed the co-localization of Mdh2p and MTS-lacking Mdh1p, we wondered whether the MTS-disrupted Mdh1p might enter peroxisomes through the hetero-oligomerization with Mdh2p/Mdh3p. To test this, we generated a yeast strain co-expressing Mdh1p(noMTS)-mCherry and Pex3p-GFP (a peroxisomal protein marker) (Fig. 5A). As shown in Fig. 5B, MTS-lacking Mdh1p appeared to co-localize with Pex3p. In a co-localization frequency analysis of MTS-lacking Mdh1p (tagged with mCherry) and Pex3p (tagged with GFP) performed on a yeast construct in the saturation phase, all cells that contained both protein signals also showed 100% co-localization. The analysis, which counted a total of 153 to 162 cells for each clone across the three experiments, revealed that the average percentage of cells containing both protein signals was 72.50% for Clone #1 and 71.58% for Clone #2. The percentages for cells showing either signal were 20.68% and 22.61% for Clone #1 and Clone #2, respectively, while the percentages of cells without any signals were 6.83% and 5.81%. The complete data for this experiment are detailed in Table S5. Combined with Fig. 3B, our findings suggest the possibility that different malate dehydrogenase isozymes can form complexes. They also suggest that an association with Mdh3p enables cytosolic Mdh2p and MTS-lacking Mdh1p to enter peroxisomes (Gabay-Maskit et al., 2020).

Our study primarily relies on fluorescence microscopy imaging to demonstrate the co-localization of MTS-lacking Mdh1p with Mdh2p and the peroxisomal marker Pex3p. While these imaging data suggest that Mdh1p lacking its mitochondrial targeting signal can associate with Mdh2p in cytoplasmic puncta and potentially enter peroxisomes, we acknowledge that co-localization alone does not definitively prove direct protein–protein interaction or peroxisomal import into the matrix. The resolution of our imaging also precludes differentiation between proteins within the peroxisomal matrix and those merely adsorbed to the membrane. Therefore, future studies will employ complementary biochemical and genetic approaches, such as co-immunoprecipitation or yeast two-hybrid assays, to confirm direct interactions between Mdh1p, Mdh2p and Mdh3p. Furthermore, to conclusively establish peroxisomal import and the proposed piggybacking mechanism via Mdh3p, experiments investigating the dependence on the peroxisomal receptor Pex5p and the consequence of forcing Mdh3p to remain cytosolic are necessary. These additional lines of evidence will provide stronger support for our hypotheses regarding the hetero-assembly and subcellular localization of malate dehydrogenase isozymes.

The TCA cycle and glyoxylate cycle are both important for the maintenance of regular metabolism and $NAD^+/NADH$ homeostasis (Bakker et al., 2001), and metabolic engineering has been gaining attention as an option to generate cells with a preferred metabolic outcome. Knowing that yeast isozymes – specifically, malate dehydrogenases – can form hetero-oligomers with each other, or engage in multi-enzyme complexes with TCA cycle, glyoxylate cycle, or even gluconeogenic enzymes, and that some of these can be re-targeted into different cellular compartments, could offer a means to control (stimulate or repress) specific biochemical reactions within given pathway(s). Based on the characterization of the different malate dehydrogenase isozymes, we propose that in specific cellular environments, Mdh3p might have a lower catalytic efficiency compared to other isozymes. According to the report by Joan S. Steffan and Lee McAlister-Henn (1992), a preliminary kinetic comparison showed that Mdh3p had a fourfold higher

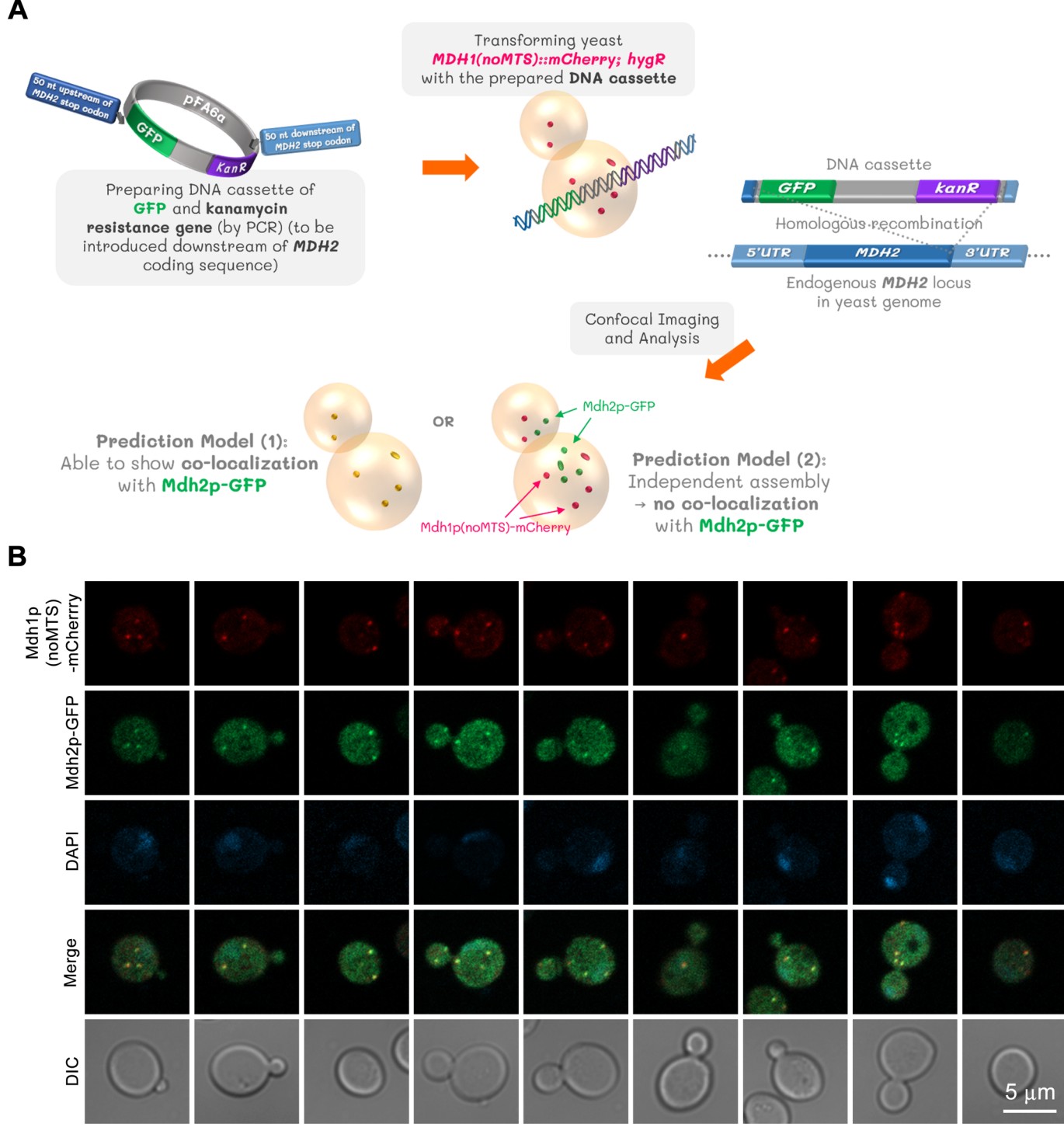

**Fig. 3. Co-localization suggests MTS-lacking Mdh1p and Mdh2p can co-assemble.** pFA6a-GFP(S65T)-kanMX6 was used as a template DNA for preparing the DNA cassette by PCR. The yeast strain expressing Mdh1p(noMTS)-mCherry was transformed with the prepared DNA cassette. The resulting yeast construct after verified to express both Mdh1p(noMTS)-mCherry and Mdh2p-GFP was subjected to imaging and analysis (A). Representative confocal images show two different yeast clones that were grown in liquid YPD to saturation (1-day cultures). The fixed cells from each yeast clone were captured as Z-stack images, which were then compressed into two-dimensional images using maximum projection. Three independent experiments were performed with similar results (B). A co-localization frequency analysis, performed on cells from the saturation phase (1-day cultures), showed that while an average of 71.27% (Clone #1) and 70.59% (Clone #2) of cells contained both protein structures, there was 100% co-localization between them. These frequencies were determined by counting a total of 152 to 164 cells for each clone. Detailed data can be found in Table S4. Mdh2p, yeast cytoplasmic malate dehydrogenase II.

apparent $K_m$ value for oxaloacetate relative to the other isozymes. They also noted that Mdh3p contributed approximately 10% of the total cellular malate dehydrogenase activity in acetate-grown cells.

This kinetic limitation could indicate that Mdh3p alone might not be sufficient to sustain the glyoxylate pathway under certain conditions. The research team reported that both *MDH2* and *MDH3*

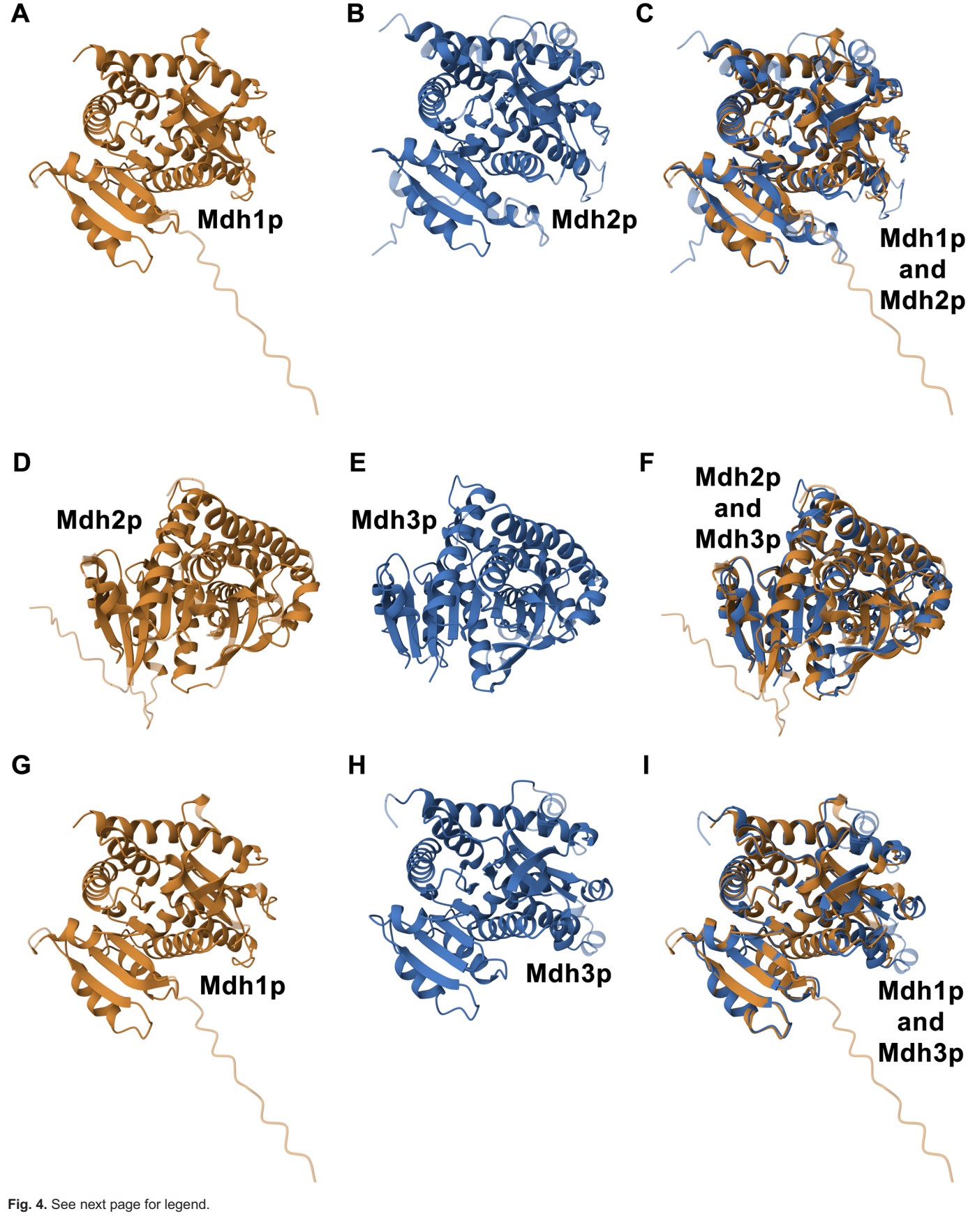

**Fig. 4.** See next page for legend.

**Fig. 4. Comparative Three-Dimensional Structures of Mdh1p, Mdh2p, and Mdh3p.** This figure presents the predicted three-dimensional structures of Mdh1p, Mdh2p, and Mdh3p, along with superimpositions highlighting their structural similarities. Panels A and G both display the AlphaFold2-predicted 3D structure of Mdh1p (UniProt ID: P17505, AF_AFP17505F1). Panels B and D show the AlphaFold2-predicted 3D structure of Mdh2p (UniProt ID: P22133, AF_AFP22133F1), while panels E and H present the AlphaFold2-predicted 3D structure of Mdh3p (UniProt ID: P32419, AF_AFP32419F1). Panels C, F, and I illustrate the structural superimpositions performed using the pairwise structure alignment tool from RCSB PDB (https://www.rcsb.org/alignment). Specifically, panel C depicts the superimposition between Mdh1p and Mdh2p (46% sequence identity), with the TM-align analysis yielding an RMSD of 1.69 and a TM Score of 0.91. Panel F shows the superimposition between Mdh2p and Mdh3p (43% sequence identity), where the TM-align analysis revealed an RMSD of 1.96 and a TM-Score of 0.84. Panel I displays the superimposition between Mdh1p and Mdh3p (50% sequence identity), with the TM-align analysis showing an RMSD of 1.52 and a TM-Score of 0.91.

disruptions cause growth defects on acetate, and they also noted that Mdh2p is associated with the peroxisomal cellular fraction. These findings indicate a functional role for both isozymes in $C_2$ carbon metabolism, supporting the idea that Mdh2p could help Mdh3p by forming a heterocomplex, thereby ensuring the continued function of the glyoxylate pathway (Steffan and McAlister-Henn, 1992). Our findings provide supporting evidence that MTS-lacking Mdh1p may co-assemble with Mdh2p. A co-localization frequency analysis showed that in an average of over 70% of cells containing both protein structures, there was 100% co-localization between them. We further showed that MTS-lacking Mdh1p can co-localize with the peroxisomal marker Pex3p. This suggests that Mdh1p, when not targeted to the mitochondria, can associate with Mdh2p and may be re-directed to the peroxisome. This information may prove valuable for future applications in medical and biotechnological fields (Easlon et al., 2008; Gibson and McAlister-Henn, 2003; Lorenz and Fink, 2001; Zelle et al., 2008; McAlister-Henn et al., 1995).

## MATERIALS AND METHODS
### Bacterial and yeast strains
*Escherichia coli* One Shot™ MAX Efficiency™ DH5α-T1R competent cells (Thermo Fisher Scientific, USA) were used for cloning and propagation of pBS35-MDH1. Bacterial cultures were maintained in LB medium [1% (w/v) Bacto™ tryptone (BD Biosciences), 0.5% (w/v) Bacto™ yeast extract (BD Biosciences), and 1% (w/v) NaCl (Merck)], supplemented with ampicillin (100 µg/ml) (PanReac AppliChem) at 37°C.

Yeast BY4741 (*MATa his3*Δ1 *leu2*Δ0 *met15*Δ0 *ura3*Δ0) (Thermo Fisher Scientific) was used as a base strain for constructing yeast *MDH1(noMTS)::mCherry; hygR*. The resulting yeast construct was subsequently used as a background strain to create another two yeast strains; (1) yeast expressing Mdh1p(noMTS)-mCherry and Mdh2p-GFP, and (2) yeast expressing Mdh1p(noMTS)-mCherry and Pex3p-GFP.

Yeast *ALD4(noMTS)::GFP; kanR*, constructed in our previous study (Noree, 2018), was used as a base strain to create yeast strain expressing Ald4p(noMTS)-GFP and Mdh1p(noMTS)-mCherry.

All yeast cultures were maintained in YPD medium [(2% (w/v) Bacto™ peptone (BD Biosciences), 1% (w/v) Bacto™ yeast extract, and 2% (w/v) glucose (Sigma-Aldrich)] at 30°C. G418 (400 µg/ml) (PanReac AppliChem) and hygromycin B (250 µg/ml) (Merck) were used for selection of the corresponding yeast strains during yeast transformation process.

### Construction of pBS35-MDH1
The coding sequence of the *MDH1* gene (stop codon excluded) was amplified by PCR using KOD Hot Start DNA Polymerase Kit (Merck). The genomic DNA (gDNA) isolated from yeast BY4741 was used as a template DNA. The PCR product was purified using GenepHlow™ Gel/PCR Kit (Geneaid), and then cloned into pBS35 (Addgene) at *Sal*I and *Sma*I recognition sites. *Sal*I and *Sma*I restriction digests were performed

according to the manufacturer's instructions (Thermo Fisher Scientific). The ligation reaction was performed using T4 DNA Ligase Kit (NEB). After bacterial transformation and selection on LB agar supplemented with ampicillin, the recombinant plasmid isolated from each bacterial transformant was extracted by using Presto™ Mini Plasmid Kit (Geneaid) for further nucleotide sequence verification by DNA sequencing (Macrogen, South Korea). The primers used for molecular cloning and DNA sequencing are shown in Table S1.

### Construction of yeast strains used for investigation in this study
We decided to engineer the yeast genome, rather than using any expression plasmids, in order to keep all the modified genes as one copy per genome by using PCR-based yeast genome engineering approach (Petracek and Longtine, 2002) with some modifications. To make yeast construct expressing Mdh1p(noMTS)-mCherry, first, the DNA cassette [carrying 50-bp upstream sequence of *MDH1* gene+start codon (ATG)+*MDH1* coding sequence without MTS sequence (nt4-51Δ)+linker sequence+mCherry coding sequence+hygromycin resistance gene+50-bp downstream sequence of *MDH1* stop codon] was prepared by PCR using pBS35-MDH1 as a template DNA. The purified DNA cassette was then transformed into yeast BY4741 using lithium acetate/polyethylene glycol/single-stranded DNA and heat-shock method. The detailed protocol for yeast transformation and selection was previously described in (Nasalingkhan et al., 2023). Some yeast transformants were randomly picked for initial screening by fluorescence microscopy. Only the clones with positive fluorescent pattern (showing mCherry signal outside mitochondria) were then chosen for further validation. To verify the successful deletion of MTS sequence from *MDH1* gene and insertion of mCherry coding sequence downstream of *MDH1*, PCR was performed using gDNA isolated from each of the selected yeast transformants as a template DNA. Three different clones of yeast *MDH1::mCherry; hygR* were also used for comparison after cell imaging. Primers were specifically designed to cover the entire *MDH1* and mCherry coding sequences. All PCR samples were then sent out for DNA sequencing.

To create yeast expressing Ald4p(noMTS)-GFP and Mdh1p(noMTS)-mCherry, the DNA cassette [carrying 50-bp upstream sequence of *MDH1* gene+start codon (ATG)+*MDH1* coding sequence without MTS sequence (nt4-51Δ)+linker sequence+mCherry coding sequence+hygromycin resistance gene+50-bp downstream sequence of *MDH1* stop codon] was transformed into yeast *ALD4(noMTS)::GFP; kanR*, instead.

To construct yeast expressing Mdh1p(noMTS)-mCherry and Mdh2p-GFP, the DNA cassette [carrying 50-bp upstream sequence of *MDH2* stop codon+linker sequence+GFP coding sequence+kanamycin resistance gene+50-bp downstream sequence of *MDH2* stop codon] was prepared by PCR using pFA6a-GFP(S65T)-kanMX6 (Addgene) as a template DNA. The purified DNA cassette was then transformed into yeast *MDH1(noMTS)::mCherry; hygR*. The resulting yeast construct was verified by both fluorescence microscopy and PCR.

Finally, to create yeast expressing Mdh1p(noMTS)-mCherry and Pex3p-GFP, the DNA cassette [carrying 50-bp upstream sequence of *PEX3* stop codon+linker sequence+GFP coding sequence+kanamycin resistance gene+50-bp downstream sequence of *PEX3* stop codon] was prepared by PCR using pFA6a-GFP(S65T)-kanMX6 as a template DNA before transforming it into yeast *MDH1(noMTS)::mCherry; hygR*. The clones showing a positive fluorescence pattern were confirmed again by PCR.

All the primers used for preparing the DNA cassettes and preparing the PCR samples for yeast strain verification by DNA sequencing, and the sequencing primers, are shown in Table S1.

### Yeast cell imaging, quantification, and statistical analysis
Yeast samples were prepared by growing them in liquid yeast rich culture medium containing yeast extract, peptone, and dextrose (YPD) at 30°C with 225 rpm shaking to the indicated incubation time before fixation with 37% (w/v) formaldehyde solution (Merck) (1 ml liquid culture and 100 µl 37% formaldehyde solution) for 15 min (in the dark with gentle agitation). After being washed (twice with 1 ml sterile water) and centrifuged at 3381 *g* at room temperature for 2 min, the fixed cells were resuspended in 1× phosphate buffered saline (PBS) (Calbiochem® OmniPur®, Merck). About 10 µl of each cell suspension was dropped onto a microscope slide

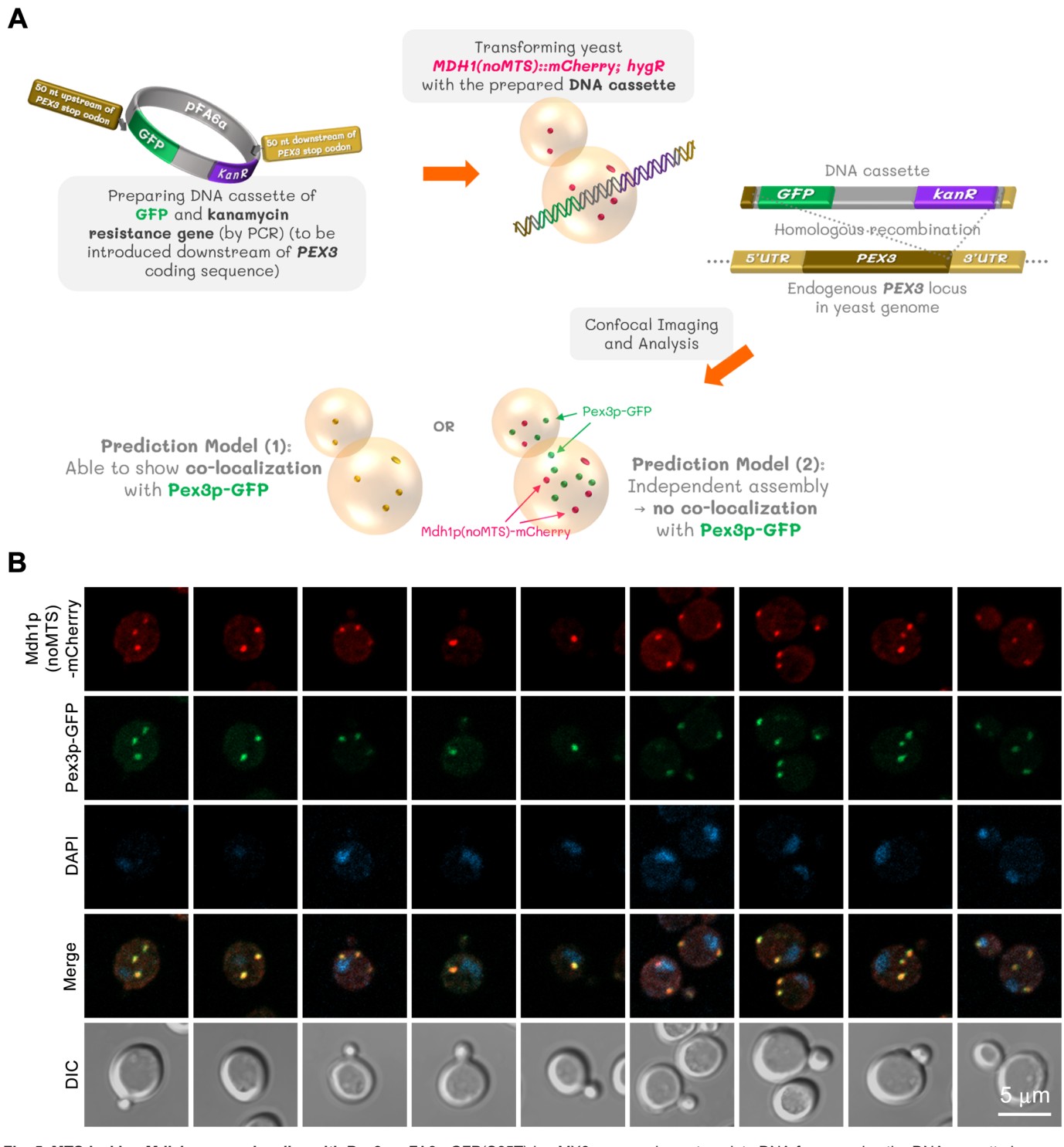

**Fig. 5. MTS-lacking Mdh1p can co-localize with Pex3p.** pFA6a-GFP(S65T)-kanMX6 was used as a template DNA for preparing the DNA cassette by PCR. The yeast strain expressing Mdh1p(noMTS)-mCherry was transformed with the prepared DNA cassette. The resulting yeast construct after verified to express both Mdh1p(noMTS)-mCherry and Pex3p-GFP was subjected to imaging and analysis (A). Representative confocal images show two different yeast clones that were grown in YPD to saturation (1-day cultures). The fixed cells from each yeast clone were captured as Z-stack images, which were then compressed into two-dimensional images with maximum projection. Three independent experiments were performed with similar results (B). A co-localization frequency analysis, performed on cells from the saturation phase (1-day cultures), showed that while an average of 72.50% (Clone #1) and 71.58% (Clone #2) of cells contained both protein signals, there was 100% co-localization between them. These frequencies were determined by counting a total of 153 to 162 cells for each clone. Detailed data can be found in Table S5. Pex3p, yeast peroxisomal membrane protein; used as a peroxisomal marker.

(Shandon SuperFrost Plus, Thermo Fisher Scientific), followed by placing a cover slip over the sample (Menzel Gläser, Thermo Fisher Scientific). The slide was then flipped over onto a lint-free lab wipe (Kimwipes, Kimtech Science) and gently pressed to remove the excess liquid and to help prevent

the cells from floating around. Nail polish was used to seal the coverslip to the prepared slide.

The yeast cells were imaged with a Zeiss LSM 800 confocal laser scanning microscope (Axio Observer.Z1) using a Plan-Apochromat

63×/1.40 oil DIC M27 objective at a magnification of 630×. Images were acquired with a size of 1024×1024 pixels, 16 bits per pixel, a scan speed of 7, and averaging of 4, in Best signal mode. The pinhole was set to 1 airy unit for each channel (mCherry: 53 μm, EGFP: 44 μm, DAPI: 38 μm). Laser filters were set at 561 nm for mCherry, 488 nm for EGFP and 405 nm for DAPI. For mCherry, the excitation wavelength was 587 nm, the emission wavelength was 610 nm, and detection was set from 570-700 nm with a detector gain of 800 V using the Airyscan detector. For EGFP, the excitation wavelength was 488 nm, the emission wavelength was 509 nm, and detection was from 450-575 nm with a detector gain of 650 V using the Airyscan detector. For DAPI, the excitation wavelength was 353 nm, the emission wavelength was 465 nm, and detection was from 400-495 nm with a detector gain of 800 V using the Airyscan detector. For DIC, detection was from 400-700 nm with a detector gain of 4 V using the Photodiode detector. Z-stack images were compressed into a single two-dimensional image with maximum projection using Zen Blue software version 2.1.57.1000.

For each counting experiment, the total number of cells was analyzed to determine several metrics: the frequency of assembly or co-localization, the percentage of cells with both structures or fluorescent signals, the percentage of co-localization between two fluorescent signals, and the percentage of cells without any structures or signals. Statistical analysis was performed by calculating the average number from three independent experiments with the standard error of the mean (SEM).

### Protein structure prediction and comparative analysis

The three-dimensional (3D) structures of *Saccharomyces cerevisiae* mitochondrial malate dehydrogenase (Mdh1p, UniProt ID: P17505), cytosolic malate dehydrogenase (Mdh2p, UniProt ID: P22133), and peroxisomal malate dehydrogenase (Mdh3p, UniProt ID: P32419) were obtained from the AlphaFold Protein Structure Database (AlphaFold 2 models: AF_AFP17505F1, AF_AFP22133F1, and AF_AFP32419F1, respectively). Structural superimpositions between protein pairs (Mdh1p and Mdh2p; Mdh2p and Mdh3p; Mdh1p and Mdh3p) were performed using the pairwise structure alignment tool available through the RCSB PDB website (https://www.rcsb.org/alignment). The alignment method employed was TM-align, which calculates both the root mean square deviation (RMSD) and the template modeling (TM) Score to quantify structural similarity.

### Acknowledgements

We thank W. Tirasophon, C. Ongvarrasopone, P. Nonejuie for kindly providing some materials and scientific instruments. We also thank the Advanced Cell Imaging Center, Institute of Molecular Biosciences, Mahidol University for confocal imaging and analysis. Some parts of this project were used for teaching MBMG615 Research Rotations in Molecular Biology (Academic year 2024, rotation students: T. Pruettijarai, M. R. Magar, and Y. Temsour). C. Chan's MSc study was partially supported by the King Prajadhipok and Queen Rambhai Barni Memorial Foundation scholarship.

### Competing interests

The authors declare no competing or financial interests.

### Author contributions

Conceptualization: C.N.; Formal analysis: C.C., C.N.; Funding acquisition: C.N.; Investigation: C.C., C.N.; Methodology: C.N.; Project administration: C.N.; Resources: C.N.; Supervision: C.N.; Validation: C.C., C.N.; Visualization: C.C., N.S., C.N.; Writing – original draft: B.K.S., J.E.W., C.N.; Writing – review & editing: B.K.S., J.E.W., C.N.

### Funding

This research project was partially supported by Mahidol University. Open Access funding provided by Mahidol University. Deposited in PMC for immediate release.

### Data and resource availability

All data generated or analyzed during this study are available within the article and its supplementary information.

### Peer review history

The peer review history is available online at https://journals.biologists.com/bio/article-lookup/doi/10.1242/bio.062199.reviewer-comments.pdf.

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
