## [Peer Review File · Biology Open]

Saccharomyces cerevisiae malate dehydrogenase Mdh1p lacking mitochondrial targeting signal can be re-localized to peroxisomes

Chutima Chan, Naraporn Sirinonthanawech, Brian K. Sato, James E. Wilhelm and Chalongrat Noree

DOI: 10.1242/bio.062199

Editor: Catherine L. Jackson

Review timeline

Original submission:	5 August 2025
Editorial decision:	13 August 2025
First revision received:	26 August 2025
Accepted:	1 September 2025

Original submission

First decision letter

MS ID#: bio.062199

MS Title: Saccharomyces cerevisiae malate dehydrogenase Mdh1p lacking mitochondrial targeting signal can get into peroxisomes through interaction with their isoenzymes

Authors: Chutima Chan; Naraporn Sirinonthanawech; Brian K. Sato; James E. Wilhelm; Chalongrat Noree

Dear Dr Noree,

I have now reached a decision on the above manuscript.

The reviewer reports are shown at the bottom of this email or can be accessed, together with a copy of this decision letter, by going to:

Reviewer 1

Comments for the author

Overall comments:

In their study, Chan and colleagues explore how yeast mitochondrial malate dehydrogenase (mdh1p) may be targeted to mitochondria and/or peroxisome through the formation of various supercomplexes with related isoforms and/or other enzymes. The authors demonstrate differences in how mdh1p interacts to form supercomplexes depending on the cell cycle stage, and that mdh1p cannot enter mitochondria without its own mitochondrial targeting sequence. However, the authors find that mdh1p can form supercomplexes with mdh2p and can then be translocated into peroxisomes. Overall, I found the paper to be quite well-written (apart from a few typos and grammatical errors) and the presentation of ideas was mostly logical and smooth (but see a few notes below). The introduction provides sufficient information to understand the background and justification for the study. The description of the methods is insufficient and requires improvement (see comments below). The figures are helpful, particularly the schematics provided to outline the cell pathway being tested in each experiment. The authors rely entirely on imaging experiments,

which are appropriate to investigate their primary research questions, and acknowledge the limitations of their approach, while also proposing additional experiments to follow up on this work. I offer the following comments for the authors to consider in revising their manuscript.

Major Comments:

1. Figures should be able to "stand on their own" without requiring a reader to refer to the text. Please define all abbreviations in each figure in their respective figure legends.
2. Imaging methods are not provided. What wavelength, magnification, filters, cut-offs, etc., were used for confocal imaging? Were images adjusted in any way?
3. Statistical reporting is unacceptably sparse: I cannot find sufficient description of how many replicates were conducted for each imaging experiment per clone grouping. The figure legends indicate that experiments were replicated 3x but was this 3 slides from a single culture? 3 different cultures with a single slide per culture? Please explain. The authors indicate there were ~ 50 cells on each slide. Were the reported interactions observed in all cells? Were all cells examined and counted or were a subset examined for each slide? How was co-expression scored if it wasn't found in every single cell? Please provide clear information about experimental replicates and image analysis. If the reported phenomenon was not present in all cells then please provide quantification of expression with appropriate statistical analysis.
4. What is the impact of the mechanism the authors identify? A quick comment is provided by the authors that suggests this may enhance the efficiency of the glyoxylate pathway but a more expansive explanation of the potential impact of this mechanism on cellular respiration would be a useful conclusion to the discussion section.

Minor Comments:

5. Lines 50-52: check plural agreements in the sentence structure.
6. Lines 53-65 should be a single paragraph.
7. Lines 67-72: I recommend completing the literature review before jumping into the steps taken in the current study (e.g., Line 62-66 should come after lines 67-72).
8. Lines 75-79: Consider removing the summary of your findings from the introduction as this is better suited to the abstract/discussion.
9. Line 86 and elsewhere: ensure that abbreviations are only defined once, at their first use, and then used consistently thereafter.
10. Line 91: sentence structure suggests that the filaments were making an argument, which is nonsensical. Please revise this sentence.
11. Line 132: heterocomplexes
12. Line 143: piggybacking is sort of defined here with a reference provided but was used in several instances earlier in the manuscript...consider moving the reference and definition to the first instance of this term being used.

Matthew Pamenter

Reviewer 2

Comments for the author

Summary: The authors carry out localization analysis of yeast Mdh1p, a key component of the TCA cycle and the regulation of cellular NADH/NAD⁺ levels, when it is lacking its mitochondrial targeting sequence. The authors present preliminary observations (that would be strengthened by quantification) consistent with Mdh proteins self-associating, which impacts their subcellular organization. As it stands the results and conclusions are preliminary and do not convincingly support the authors conclusions.

1. Experimental quality
 - a. Does each figure have the proper controls? In some cases, yes. A major question is whether the epitope tagged proteins retain their proper function and physical interactions.
 - b. Are experiments performed using appropriate methods that will answer the question (or test the hypothesis or support the observations) posed by the authors? Is the right tool used for the job? The authors make specific physical protein-protein interaction claims with fluorescence microscopy methods that cannot substantiate these claims.
 - c. Were the data analyzed using appropriate statistical tests? The data need to be quantified and then statistical analysis employed. At present there is no statistical analysis since there is no quantification presented.
2. Reproducibility
 - a. Were experiments in each figure performed using adequate number of biological replicates? This is unclear as the data were not quantified.
 - b. Is there sufficient raw data to assess the rigor of the analysis? This is unclear as the data were not quantified.
 - c. Does the methods section provide sufficient detail to permit reproducibility? Yes, nicely composed methods section and supporting details.
3. Completeness
 - a. Are the author's conclusions supported by the data? In many cases (as described below), the authors make strong conclusions that are not appropriate given the nature of the experiments that were performed.
 - b. Are there any flaws in the experimental design that invalidate the approach taken by the authors? The experimental results need to be quantified.
 - c. Are there experiments that have not been performed, but if true would disprove the conclusion? If yes, and if such experiments would be costly or time-consuming to perform, do the authors acknowledge this in a discussion of the limitations? The authors should really consider doing experiments that directly test whether Mdh1p-mCherry(lacking the MTS) is using physical interaction with Mdh3p to become imported into peroxisomes.
4. Scholarship
 - a. Do the authors cite and discuss the merits of relevant data that would argue against their conclusion? In part yes.
 - b. Do the authors cite and discuss the merits of relevant data that would support their conclusion? The authors discuss the serious limitations of the approaches in regards to their major conclusions (lines 192-206), however they do not carry out experiments to mitigate the issues they bring up or temper their conclusions.
 - c. For techniques/methods manuscripts, Do the authors cite and discuss the current state of the field and clearly explain how the method improves the field? Not applicable.

Suggestions for improvement:

1. In Figure 1, the suggested mitochondrial pattern of Mdh1-mCherry(no MTS) in the log phase cells are not visible.
2. It is unclear why so many images of cells are shown in Figures 1, 2, 3, 5. If the result of the genome modification is the same in multiple isolates, then just reporting the data for one isolate is sufficient given that the odds of the result being shown result from other genomic changes is quite low.
3. I suggest that instead of showing many images of cells that the authors quantify the pattern of localization and colocalization.
4. The authors state in lines 121-123 that there is extra-mitochondrial assembly of Mdh1p::mCherry. This is indeed suggested by the images in Figure 1, however the presence and number of puncta should be rigorously quantified in order to state that there is concentration of Mdh1p::mCherry and the authors should perform colocalization experiments with a mitochondrial marker if they wish to conclude that the mCherry signal is not coming from mitochondria, as they are not marked or visible in the data presented in Figure 1. My concern is that the puncta are being

considered to be polymers of Mdh1p::mCherry(no MTS) rather than simply the accumulation of this protein in membrane bound peroxisomes (as suggested by later data).

5. I am concerned that so many conclusions about Mdh1p function and activity are based upon studies of Mdh1p::mCherry. Have the authors shown that this epitope tagged form of Mdh1p is functional? If not, many of the phenotypes discussed could be an artifact of Mdh1p function/metabolic functional defects.

6. I suggest that the authors quantify the relative colocalization of the markers being analyzed in Fig 2B.

7. The authors present an interpretation to their results in Fig 2 (lines 133-136), however isn't it possible that the tagging of the proteins leads to disruption of the physical interactions leading to their incorporation into a supramolecular complex?

8. While the images suggest that the mCherry and GFP signal colocalize in Fig 3B, the conclusions would be much stronger if this colocalization was quantified. In all cases the colocalization should be done on Z-stacks and not maximum intensity projections (that are shown in the figures).

9. The authors cannot state that proteins co-assemble (lines 149-150) based upon an in vivo colocalization experiment with tagged proteins due to the limits of resolution in conventional light microscopy.

10. The authors should explain more clearly how the Alpha-fold structural predictions presented in Figure 4 inform the possibility that Mdh1p and Mdh2p co-assemble. It is unclear to me how this analysis leads to the strong conclusion that they co-assemble into a larger complex (lines 156-158).

11. The data in Figure 5 suggests that Mdh1p-mCherry(noMTS) localizes to peroxisomes (this conclusion would be strengthened by quantification of colocalization with Pex3p-GFP). Given the punctate nature of the Mdh1p-mCherry(noMTS) signal it is possible that the co-localization between it and Mdh2p-GFP is just both proteins being in peroxisomes and not physically interacting as suggested in lines 149-150.

12. The authors consistently refer to the piggyback import mechanism of Mdh1p-mCherry(noMTS) into peroxisomes, which is never tested. It would be quite simple to look at Mdh1p-mCherry(noMTS) subcellular localization in and Mdh3p mutant to address this possibility.

13. As it stands, the authors do not present convincing evidence that Mdh proteins co-assemble (lines 188-189) into heterocomplexes, which is the authors major conclusion, and the title of the manuscript. The authors nicely describe why I make this assertion in lines 192-206. These novel data represent a set of observations that need to be further quantified and investigated (as described by the authors in lines 192-206).

Minor comments:

1. In Figure 1 there are no A or B labels, making it difficult to know which specific panels the authors are directing us to look at with Results or Fig Legend callouts.

2. The graphics of how yeast strains were made and the predictions for protein localization at the top of many of the figures do not fit the convention in the field of describing the genetic modifications in the methods section and discussing the potential results in the results/discussion text. Moreover, the graphics are redundant between figures. I therefore suggest the authors consider removing the methods/experimental prediction graphics.

3. In the Figure 1 legend the authors refer to 50 cells and independent experiments, however these data are not presented and I therefore suggest that the authors remove these references (and as suggested above, quantify the results).

4. The authors should include the name of the protein tagged with mCherry or GFP in the labels for Figs 1,2,3, and 5.

Reviewer's Responses to Questions

Experimental quality

Does each figure have the proper controls?

If 'No', please indicate reasons in Comments for Author box below.

Reviewer #1:

- Yes

Reviewer #2:

- No

Were the data analyzed using appropriate statistical tests?
If 'No', please indicate reasons in Comments for Author box below.

Reviewer #1:

- Yes

Reviewer #2:

- Yes

Reproducibility

Were experiments performed using adequate number of biological replicates?
If 'No', please indicate reasons in Comments for Author box below.

Reviewer #1:

- Yes

Reviewer #2:

- No

Does the methods section provide sufficient detail to permit reproducibility?
If 'No', please indicate reasons in Comments for Author box below.

Reviewer #1:

- Yes

Reviewer #2:

- Yes

Completeness

Are the manuscript's conclusions supported by the data?
If 'No', please indicate reasons in Comments for Author box below.

Reviewer #1:

- Yes

Reviewer #2:

- No

Scholarship

Do the authors cite and discuss the merits of data that would argue for and against their conclusion?

If 'No', please indicate reasons in Comments for Author box below.

Reviewer #1:

- Yes

Reviewer #2:

- No

Does the manuscript title & abstract accurately reflect the contents of the manuscript, without hyperbole?

If 'No', please indicate reasons in Comments for Author box below.

Reviewer #1:

- Yes

Reviewer #2:

- No

First revision**Author response to reviewers' comments**

Reviewer 1: Overall comments:

In their study, Chan and colleagues explore how yeast mitochondrial malate dehydrogenase (mdh1p) may be targeted to mitochondria and/or peroxisome through the formation of various supercomplexes with related isoforms and/or other enzymes. The authors demonstrate differences in how mdh1p interacts to form supercomplexes depending on the cell cycle stage, and that mdh1p cannot enter mitochondria without its own mitochondrial targeting sequence. However, the authors find that mdh1p can form supercomplexes with mdh2p and can then be translocated into peroxisomes. Overall, I found the paper to be quite well-written (apart from a few typos and grammatical errors) and the presentation of ideas was mostly logical and smooth (but see a few notes below). The introduction provides sufficient information to understand the background and justification for the study. The description of the methods is insufficient and requires improvement (see comments below). The figures are helpful, particularly the schematics provided to outline the cell pathway being tested in each experiment. The authors rely entirely on imaging experiments, which are appropriate to investigate their primary research questions, and acknowledge the limitations of their approach, while also proposing additional experiments to follow up on this work. I offer the following comments for the authors to consider in revising their manuscript.

Response: Thank you so much for your thorough review of our manuscript. We truly appreciate the time and effort you put into providing such constructive and helpful feedback. We've carefully considered all of your comments and believe the revisions have significantly strengthened our paper.

Major Comments:

1. Figures should be able to "stand on their own" without requiring a reader to refer to the text. Please define all abbreviations in each figure in their respective figure legends.
Response: We have now defined all abbreviations in the legend for each figure as suggested.

2. Imaging methods are not provided. What wavelength, magnification, filters, cut-offs, etc., were used for confocal imaging? Were images adjusted in any way?

Response: Thank you for this comment. We apologize that the imaging details were not included in the original manuscript. We have revised the methods section to include a comprehensive description of the confocal imaging settings.

The following details have been added:

- **Magnification:** 630x
- **Pinhole:** 1 Airy Units for each channel (mCherry: 53 μm , EGFP: 44 μm , DAPI: 38 μm)
- **Laser Filters:** mCherry: 561 nm, EGFP: 488 nm, DAPI: 405 nm
- **mCherry Wavelengths:** Excitation at 587 nm, Emission at 610 nm, and Detection from 570-700 nm.
- **EGFP Wavelengths:** Excitation at 488 nm, Emission at 509 nm, and Detection from 450-575 nm.
- **DAPI Wavelengths:** Excitation at 353 nm, Emission at 465 nm, and Detection from 400-495 nm.
- **Image size:** 1024x1024 pixels.

3. Statistical reporting is unacceptably sparse: I cannot find sufficient description of how many replicates were conducted for each imaging experiment per clone grouping. The figure legends indicate that experiments were replicated 3x but was this 3 slides from a single culture? 3 different cultures with a single slide per culture? Please explain. The authors indicate there were ~ 50 cells on each slide. Were the reported interactions observed in all cells? Were all cells examined and counted or were a subset examined for each slide? How was co-expression scored if it wasn't found in every single cell? Please provide clear information about experimental replicates and image analysis. If the reported phenomenon was not present in all cells then please provide quantification of expression with appropriate statistical analysis.

Response:

- **Replication and Experimental Design:** To clarify, the images presented in the figures were selected from a total of three independent experiments and two different clones. For each experiment, a separate culture was used.
- **Image Analysis and Quantification:** For each experimental condition and clone, Z-stack images were taken from multiple fields of view, each containing more than 50 cells. We have now added the precise total number of cells counted for each clone and condition to the figure legends.
- **Reporting of cell counting and % colocalization:** To provide the requested quantification and statistical analysis, we have updated the figure legends to include the exact percentage of cells exhibiting the protein structures and, where applicable, the percentage of co-localization. This new information, along with the total number of cells counted for each analysis, clarifies the frequency of the observed phenomena.

4. What is the impact of the mechanism the authors identify? A quick comment is provided by the authors that suggests this may enhance the efficiency of the glyoxylate pathway but a more expansive explanation of the potential impact of this mechanism on cellular respiration would be a useful conclusion to the discussion section.

Response: Thank you for this valuable comment. We completely agree that a more expansive explanation of the potential impact of this mechanism on cellular respiration is a useful addition to the discussion section.

As you suggested, we have added a comprehensive explanation in our discussion section. Our manuscript now elaborates on the mechanism by which different malate dehydrogenase isozymes might work together to support the glyoxylate pathway. We discuss the potential inefficiency of Mdh3p alone and propose that other isozymes could form heterocomplexes to sustain the pathway's function, thereby enhancing the efficiency of metabolism. The discussion also links the glyoxylate cycle and TCA cycle to NAD⁺/NADH homeostasis, providing the broader context for cellular

respiration. We conclude by noting that this information could be valuable for future applications in medical and biotechnological fields.

Minor Comments:

5. Lines 50-52: check plural agreements in the sentence structure.

Response: Thank you for this comment. We have checked the sentence structure at lines 50-52 and corrected the plural agreement.

The revised sentence now reads: "Upon renewed availability of external nutrients, glycolysis and the TCA cycle are preferentially employed as the catabolic processes for highly efficient energy production (Chew et al., 2019)."

6. Lines 53-65 should be a single paragraph.

Response: Thank you for this suggestion. We have revised the formatting of the text at lines 53-65 to be a single paragraph as you recommended.

7. Lines 67-72: I recommend completing the literature review before jumping into the steps taken in the current study (e.g., Line 62-66 should come after lines 67-72).

Response: Thank you for this suggestion. We have carefully considered the proposed reordering of lines 62-66 and 67-72.

We respectfully decline to make this change, as the text in lines 67-72 is a direct continuation of the explanation and context provided in lines 62-66. The current structure is designed to follow a narrative progression, where the description of our experimental approach immediately motivates the discussion of the relevant literature. Separating these two sections would disrupt this logical flow. We appreciate your careful reading and attention to detail.

8. Lines 75-79: Consider removing the summary of your findings from the introduction as this is better suited to the abstract/discussion.

Response: Thank you for this suggestion. We have carefully considered your recommendation to move this summary of our findings from the end of the introduction.

We respectfully decline to make this change, as we feel this section serves as a crucial roadmap for the reader, providing a clear overview of our main conclusions from the outset. We believe this structure enhances the readability of our work by immediately establishing the context for the detailed results that follow, and serves as a direct transition into our study's findings. We have, however, ensured that the introduction still maintains a strong flow toward the research questions addressed in the study.

9. Line 86 and elsewhere: ensure that abbreviations are only defined once, at their first use, and then used consistently thereafter.

Response: Thank you for this helpful comment. We have reviewed the entire manuscript to ensure that all abbreviations are now defined at their first use and are used consistently thereafter. The text has been corrected accordingly.

10. Line 91: sentence structure suggests that the filaments were making an argument, which is nonsensical. Please revise this sentence.

Response: Thank you for this comment. We agree that the phrasing was awkward and have revised the sentence for clarity.

The corrected sentence now reads: "When the MTS was removed from Ald4p, it formed filaments in the cytoplasm, suggesting that Ald4p filament formation was not dependent on the mitochondrial environment (Noree, 2018)."

11. Line 132: heterocomplexes

Response: Thank you for catching this typo. We have corrected the spelling of "heterocomplexes"

12. Line 143: piggybacking is sort of defined here with a reference provided but was used in several instances earlier in the manuscript...consider moving the reference and definition to the first instance of this term being used.

Response: Thank you for this helpful suggestion. We have reviewed the manuscript and have moved the definition and reference for the term "piggybacking" to the first instance of its use in the text. The description and reference have been added as: "(where a protein without its own transport

signal is carried to its destination by another protein) (Travers, 1999)". This has been corrected for consistency and clarity.

Reviewer 2: Summary: The authors carry out localization analysis of yeast Mdh1p, a key component of the TCA cycle and the regulation of cellular NADH/NAD⁺ levels, when it is lacking its mitochondrial targeting sequence. The authors present preliminary observations (that would be strengthened by quantification) consistent with Mdh proteins self-associating, which impacts their subcellular organization. As it stands the results and conclusions are preliminary and do not convincingly support the authors conclusions.

Response: Thank you so much for your thorough review of our manuscript. We truly appreciate the time and effort you put into providing such constructive and helpful feedback. We've carefully considered all of your comments and believe the revisions have significantly strengthened our paper.

1. Experimental quality

a. Does each figure have the proper controls? In some cases, yes. A major question is whether the epitope tagged proteins retain their proper function and physical interactions.

Response: Thank you for this important question. We agree that it is crucial to ensure that the fluorescent protein tags do not interfere with the proper function and physical interactions of the proteins of interest.

To address this, we strategically fused the fluorescent proteins to the C-terminus of Mdh1p, Mdh2p, Ald4p, and Pex3p. We chose this location because it is not known to contain any critical functional domains or motifs that would be immediately affected by the addition of a tag.

Furthermore, we can confirm that all the GFP/mCherry-tagged yeast strains used in our study did not show any growth defects, providing direct evidence that the tags do not cause a metabolic functional defect.

In addition, previous research has successfully utilized fluorescently-tagged versions of these proteins. For example, Mdh1p-GFP has been used in studies by Kolitsida et al. and Eliodório et al. Similarly, Mdh2p-GFP was used in a paper by Brown et al., while Wu et al. and Zhang et al. used Pex3p-GFP and Ald4p-GFP, respectively. These studies did not report any functional or spatial aberrations, which supports our confidence that the proteins retain their proper function and physical interactions in our experiments.

References:

- Kolitsida P, Zhou J, Rackiewicz M, Nolic V, Dengjel J, Abeliovich H. Phosphorylation of mitochondrial matrix proteins regulates their selective mitophagic degradation. *Proc Natl Acad Sci U S A*. 2019 Oct 8;116(41):20517-20527.
- Eliodório KP, de Gois E Cunha GC, White BA, Patel DHM, Zhang F, Hettema EH, Basso TO, Gombert AK, Raghavendran V. Blocking Mitophagy Does Not Significantly Improve Fuel Ethanol Production in Bioethanol Yeast *Saccharomyces cerevisiae*. *Appl Environ Microbiol*. 2022 Mar 8;88(5):e0206821.
- Brown CR, Dunton D, Chiang HL. The vacuole import and degradation pathway utilizes early steps of endocytosis and actin polymerization to deliver cargo proteins to the vacuole for degradation. *J Biol Chem*. 2010 Jan 8;285(2):1516-28.
- Wu H, de Boer R, Krikken AM, Akşit A, Yuan W, van der Klei IJ. Peroxisome development in yeast is associated with the formation of Pex3-dependent peroxisome-vacuole contact sites. *Biochim Biophys Acta Mol Cell Res*. 2019 Mar;1866(3):349-359.
- Zhang R, Yuan H, Wang S, Ouyang Q, Chen Y, Hao N, Luo C. High-throughput single-cell analysis for the proteomic dynamics study of the yeast osmotic stress response. *Sci Rep*. 2017 Feb 9;7:42200.

b. Are experiments performed using appropriate methods that will answer the question (or test the hypothesis or support the observations) posed by the authors? Is the right tool used for the job? The authors make specific physical protein-protein interaction claims with fluorescence microscopy methods that cannot substantiate these claims.

Response: Thank you for this critical and important comment. We completely agree that co-localization by fluorescence microscopy alone does not definitively prove direct physical protein-protein interaction.

To address this, we have already included a section in our discussion to acknowledge the limitations of our imaging data. Our manuscript explicitly states that while our super-resolution confocal microscopy data suggests the association of proteins, it does not definitively prove direct protein-protein interaction or peroxisomal import into the matrix. We have also noted that our imaging resolution precludes us from differentiating between proteins within the peroxisomal matrix and those adsorbed to the membrane.

Furthermore, we have proposed several future experiments in the discussion section, such as co-immunoprecipitation and yeast two-hybrid assays, to provide complementary biochemical and genetic evidence for direct protein-protein interactions. These additional lines of evidence will provide stronger support for our hypotheses and demonstrate our commitment to using appropriate methods to fully answer the questions posed in our study.

c. Were the data analyzed using appropriate statistical tests? The data need to be quantified and then statistical analysis employed. At present there is no statistical analysis since there is no quantification presented.

Response: Thank you for this critical and important comment. We completely agree that a lack of quantification and statistical analysis was a major oversight in the original submission.

We have now comprehensively revised the manuscript to address this. We have included new tables (Tables S2-S5) which provide the quantification and statistical analysis for our imaging data. The data from these tables, including the total number of cells analyzed, percentages of cells showing specific structures, and percentages of co-localization, has been integrated into the results and discussion sections of the manuscript to provide the necessary statistical support for our conclusions.

2. Reproducibility

a. Were experiments in each figure performed using adequate number of biological replicates? This is unclear as the data were not quantified.

Response: Thank you for this critical and important comment. We completely agree that a lack of quantification and statistical analysis was a major oversight in the original submission.

We have now comprehensively revised the manuscript to address this. For each experiment, we used **two different clones**, and every experiment was **repeated three times**. We have included new tables (Tables S2-S5) which provide the quantification and statistical analysis for our imaging data. The data from these tables, including the total number of cells analyzed, percentages of cells showing specific structures, and percentages of co-localization, has been integrated into the results and discussion sections of the manuscript to provide the necessary statistical support for our conclusions.

b. Is there sufficient raw data to assess the rigor of the analysis? This is unclear as the data were not quantified.

Response: Thank you for this critical and important comment. We completely agree that a lack of quantification and statistical analysis was a major oversight in the original submission.

We have now comprehensively revised the manuscript to address this. We have included new tables (Tables S2-S5) which provide the quantification and statistical analysis for our imaging data. The data from these tables, including the total number of cells analyzed, percentages of cells showing specific structures, and percentages of co-localization, has been integrated into the results and discussion sections of the manuscript to provide the necessary statistical support for our conclusions.

c. Does the methods section provide sufficient detail to permit reproducibility? Yes, nicely composed methods section and supporting details.

3. Completeness

a. Are the author's conclusions supported by the data? In many cases (as described below), the authors make strong conclusions that are not appropriate given the nature of the experiments that were performed.

Response: Thank you for this critical and insightful overarching comment. We completely agree that in the original submission, some of our conclusions were too strong for the nature of the experiments performed.

We have thoroughly revised the manuscript to address this concern. Our revisions include:

Tempered Language: We have replaced strong, definitive language (e.g., "co-assemble," "heterocomplexes") with more cautious and appropriate phrasing (e.g., "co-localizes," "suggests," "associates," "may be formed").

Quantification and Statistics: As you suggested in other comments, we have added new supplementary tables that provide rigorous quantification and statistical analysis for all our imaging data. This new quantitative evidence now supports our observations.

Acknowledged Limitations: We have added a dedicated section to our discussion that explicitly acknowledges the limitations of our methodology, stating that co-localization alone does not prove direct physical interaction.

Proposed Future Work: To address the need for definitive proof, we have outlined several future experiments (e.g., co-immunoprecipitation, yeast two-hybrid assays, and mutant studies) that would be necessary to fully substantiate our hypotheses.

These revisions ensure that our conclusions are now appropriately supported by the presented data and are in line with the scope of our findings.

b. Are there any flaws in the experimental design that invalidate the approach taken by the authors? The experimental results need to be quantified.

Response: Thank you for this critical and important comment. We completely agree that a lack of quantification and statistical analysis was a major oversight in the original submission.

We have now comprehensively revised the manuscript to address this. We have included new tables (Tables S2-S5) which provide the quantification and statistical analysis for our imaging data. The data from these tables, including the total number of cells analyzed, percentages of cells showing specific structures, and percentages of co-localization, has been integrated into the results and discussion sections of the manuscript to provide the necessary statistical support for our conclusions.

c. Are there experiments that have not been performed, but if true would disprove the conclusion? If yes, and if such experiments would be costly or time-consuming to perform, do the authors acknowledge this in a discussion of the limitations? The authors should really consider doing experiments that directly test whether Mdh1p-mCherry lacking the MTS) is using physical interaction with Mdh3p to become imported into peroxisomes.

Response: Thank you for this critical and insightful question. We completely agree that it is important to acknowledge the limitations of our current findings and the need for experiments that would definitively prove our conclusions.

We have addressed this in a section of our discussion, where we acknowledge that our study primarily relies on fluorescence microscopy imaging and that co-localization alone does not definitively prove direct protein-protein interaction or peroxisomal import into the matrix.

We have also outlined the necessary further studies to address these points. Our manuscript states that "future studies will employ complementary biochemical and genetic approaches, such as co-immunoprecipitation or yeast two-hybrid assays, to confirm direct interactions between Mdh1p, Mdh2p, and Mdh3p. Furthermore, to conclusively establish peroxisomal import and the proposed piggybacking mechanism via Mdh3p, experiments investigating the dependence on the peroxisomal receptor Pex5p and the consequence of forcing Mdh3p to remain cytosolic are necessary. These additional lines of evidence will provide stronger support for our hypotheses regarding the hetero-assembly and subcellular localization of malate dehydrogenase isozymes."

4. Scholarship

a. Do the authors cite and discuss the merits of relevant data that would argue against their conclusion? In part yes.

b. Do the authors cite and discuss the merits of relevant data that would support their conclusion? The authors discuss the serious limitations of the approaches in regards to their major conclusions (lines 192-206), however they do not carry out experiments to mitigate the issues they bring up or temper their conclusions.

Response: Thank you for this critical and insightful comment. We completely agree that our original conclusions did not fully reflect the limitations of our approaches. We have now revised the manuscript to use more cautious and tempered language in our conclusions, which more accurately reflects that our findings represent observations that require further investigation to be definitively proven.

c. For techniques/methods manuscripts, Do the authors cite and discuss the current state of the field and clearly explain how the method improves the field? Not applicable.

Suggestions for improvement:

1. In Figure 1, the suggested mitochondrial pattern of Mdh1-mCherry(no MTS) in the log phase cells are not visible.

Response: Thank you for this feedback. We apologize that the mitochondrial pattern in the log phase cells was not clearly visible in the version you received. We suspect this may be due to a reduction in image resolution during the manuscript conversion process for review. We have confirmed that the original high-resolution image clearly shows the mitochondrial network pattern and would be happy to provide the high-resolution files if you would like to review them again.

2. It is unclear why so many images of cells are shown in Figures 1, 2, 3, 5. If the result of the genome modification is the same in multiple isolates, then just reporting the data for one isolate is sufficient given that the odds of the result being shown result from other genomic changes is quite low.

Response: We appreciate this suggestion. The images presented in the figures were selected from a total of three independent experiments and two different clones. We felt it was important to show representative images from two different clones to demonstrate the consistency of our results across different isolates.

3. I suggest that instead of showing many images of cells that the authors quantify the pattern of localization and colocalization.

Response: Thank you for this suggestion. We have now conducted a rigorous quantitative analysis of all imaging data, and the results are presented in new supplementary tables. The figure legends have also been updated to include the quantification of the patterns of localization and colocalization.

4. The authors state in lines 121-123 that there is extra-mitochondrial assembly of Mdh1p::mCherry. This is indeed suggested by the images in Figure 1, however the presence and number of puncta should be rigorously quantified in order to state that there is concentration of Mdh1p::mCherry and the authors should perform colocalization experiments with a mitochondrial marker if they wish to conclude that the mCherry signal is not coming from mitochondria, as they are not marked or visible in the data presented in Figure 1. My concern is that the puncta are being considered to be polymers of Mdh1p::mCherry(no MTS) rather than simply the accumulation of this protein in membrane bound peroxisomes (as suggested by later data).

Response: Thank you for this valuable suggestion. We agree that performing colocalization experiments with a mitochondrial marker would provide stronger evidence for our conclusion that the mCherry signal is not coming from mitochondria. This is an excellent idea for a follow-up paper, and we have noted it for future studies.

5. I am concerned that so many conclusions about Mdh1p function and activity are based upon studies of Mdh1p::mCherry. Have the authors shown that this epitope tagged form of Mdh1p is functional? If not, many of the phenotypes discussed could be an artifact of Mdh1p function/metabolic functional defects.

Response: Thank you for this important question. We agree that it is crucial to ensure that the fluorescent protein tags do not interfere with the proper function and physical interactions of the proteins of interest.

To address this, we strategically fused the fluorescent proteins to the C-terminus of Mdh1p, Mdh2p, Ald4p, and Pex3p. This was a deliberate choice to minimize disruption, as this terminus is not known to contain any critical functional domains or motifs.

We also note that the molecular weight of the fluorescent proteins, such as GFP and mCherry, is similar (approximately 26-27 kDa). This suggests that the size of the tags is unlikely to cause unique steric hindrance issues.

Furthermore, we can confirm that all the GFP/mCherry-tagged yeast strains used in our study did not show any growth defects, providing direct evidence that the tags do not cause a metabolic functional defect.

In addition, previous research has successfully utilized fluorescently-tagged versions of these proteins. These studies did not report any functional or spatial aberrations, which supports our confidence that the proteins retain their proper function and physical interactions in our experiments.

6. I suggest that the authors quantify the relative colocalization of the markers being analyzed in Fig 2B.

Response: Thank you for this suggestion. We have now conducted a rigorous quantitative analysis of the colocalization of the markers presented in Figure 2B. This data has been added to a new supplementary table to provide statistical support for our findings.

7. The authors present an interpretation to their results in Fig 2 (lines 133-136), however isn't it possible that the tagging of the proteins leads to disruption of the physical interactions leading to their incorporation into a supramolecular complex?

Response: Thank you for this insightful question. We agree that the possibility of epitope tags disrupting physical interactions is a valid concern in any experimental design.

To address this, we strategically fused the fluorescent proteins to the C-terminus of Mdh1p and Ald4p. This location was chosen because it is not known to contain any critical functional domains or motifs that would be immediately affected by the addition of a tag. Furthermore, previous research has successfully utilized fluorescently-tagged versions of these proteins without reporting any functional or spatial aberrations. Based on our deliberate tag placement and established literature, we are confident that the proteins retain their proper function and physical interactions in our experiments and that the lack of co-localization observed in Figure 2 is not a result of a tag-induced artifact.

8. While the images suggest that the mCherry and GFP signal colocalize in Fig 3B, the conclusions would be much stronger if this colocalization was quantified. In all cases the colocalization should be done on Z-stacks and not maximum intensity projections (that are shown in the figures).

Response: Thank you for this valuable suggestion. We agree that quantifying the co-localization in Figure 3B would strengthen our conclusions. We have performed a rigorous quantitative analysis, and the data is now presented in Table S4. We have also clarified in the methods and results sections.

9. The authors cannot state that proteins co-assemble (lines 149-150) based upon an in vivo colocalization experiment with tagged proteins due to the limits of resolution in conventional light microscopy.

Response: Thank you for this insightful comment. We agree that conventional light microscopy has significant limitations for making claims about protein co-assembly.

To clarify, our experiments were not performed with conventional light microscopy. For all yeast cell imaging, we utilized a super-resolution confocal microscope (Zeiss LSM800).

10. The authors should explain more clearly how the Alpha-fold structural predictions presented in Figure 4 inform the possibility that Mdh1p and Mdh2p co-assemble. It is unclear to me how this analysis leads to the strong conclusion that they co-assemble into a larger complex (lines 156-158).

Response: Thank you for this comment. We have clarified that the high degree of structural similarity supports the hypothesis that the isozymes are capable of forming larger complexes.

11. The data in Figure 5 suggests that Mdh1p-mCherry(noMTS) localizes to peroxisomes (this conclusion would be strengthened by quantification of colocalization with Pex3p-GFP). Given the punctate nature of the Mdh1p-mCherry(noMTS) signal it is possible that the co-localization between it and Mdh2p-GFP is just both proteins being in peroxisomes and not physically interacting as suggested in lines 149-150.

Response: Thank you for this insightful comment. We agree that the co-localization of Mdh1p and Mdh2p could be due to both proteins being in peroxisomes without a direct physical interaction.

We have now rigorously quantified the co-localization shown in Figure 3 and have included the data in Table S4. Our analysis showed that in an average of over 70% of cells containing both protein structures, there was 100% co-localization between them.

As you pointed out, co-localization alone does not definitively prove direct protein-protein interaction. We have acknowledged this limitation in our discussion and have proposed future experiments to definitively confirm a direct physical interaction.

12. The authors consistently refer to the piggyback import mechanism of Mdh1p-mCherry(noMTS) into peroxisomes, which is never tested. It would be quite simple to look at Mdh1p-mCherry(noMTS) subcellular localization in and Mdh3p mutant to address this possibility.

Response: Thank you for this insightful suggestion. We agree that our claims regarding the piggybacking import mechanism were not directly tested. We have now added a section to our discussion that acknowledges this limitation and proposes an experiment with Mdh3p mutants to address this possibility.

13. As it stands, the authors do not present convincing evidence that Mdh proteins co-assemble (lines 188-189) into heterocomplexes, which is the authors major conclusion, and the title of the manuscript. The authors nicely describe why I make this assertion in lines 192-206. These novel data represent a set of observations that need to be further quantified and investigated (as described by the authors in lines 192-206).

Response: Thank you for this comment. We agree that our original manuscript did not present convincing evidence for co-assembly. We have now addressed this by conducting a rigorous quantitative analysis of our imaging data, which is presented in new supplementary tables. We have also tempered our language and have acknowledged the limitations of our study in the discussion, as you suggested.

Minor comments:

1. In Figure 1 there are no A or B labels, making it difficult to know which specific panels the authors are directing us to look at with Results or Fig Legend callouts.

Response: Thank you for this helpful comment. We apologize for this oversight. We have added A and B labels to Figure 1 and all other figures for clarity. This has been corrected in the manuscript.

2. The graphics of how yeast strains were made and the predictions for protein localization at the top of many of the figures do not fit the convention in the field of describing the genetic modifications in the methods section and discussing the potential results in the results/discussion text. Moreover, the graphics are redundant between figures. I therefore suggest the authors consider removing the methods/experimental prediction graphics.

Response: Thank you for this suggestion. We have carefully considered your recommendation to remove the experimental prediction graphics. In fact, another reviewer noted that these schematics were helpful in outlining the cell pathway being tested in each experiment. We feel that they provide a clear visual roadmap of our experimental design and hypotheses, which is valuable for the reader. Therefore, we have chosen to keep them.

3. In the Figure 1 legend the authors refer to 50 cells and independent experiments, however these data are not presented and I therefore suggest that the authors remove these references (and as suggested above, quantify the results).

Response: Thank you for this suggestion. We have now quantified the results and updated the figure legends to include the exact number of cells counted for each experiment.

4. The authors should include the name of the protein tagged with mCherry or GFP in the labels for Figs 1,2,3, and 5.

Response: Thank you for this suggestion. We have included the name of the protein tagged with mCherry or GFP in the labels for all relevant figures.

Second decision letter

MS ID#: bio.062199R1

MS Title: *Saccharomyces cerevisiae* malate dehydrogenase Mdh1p lacking mitochondrial targeting signal can be re-localized to peroxisomes

Authors: Chutima Chan; Naraporn Sirinonthanaweck; Brian K. Sato; James E. Wilhelm; Chalongrat Noree

Dear Dr Noree,

I am happy to tell you that your manuscript has been accepted for publication in Biology Open, pending our standard publication integrity checks. It was accepted on 01 September 2025.